# Deep Reinforcement Learning for Physical Layer Security Enhancement in Energy Harvesting Based Cognitive Radio Networks

**DOI:** 10.3390/s23020807

**Published:** 2023-01-10

**Authors:** Ruiquan Lin, Hangding Qiu, Weibin Jiang, Zhenglong Jiang, Zhili Li, Jun Wang

**Affiliations:** 1College of Electrical Engineering and Automation, Fuzhou University, Fuzhou 350108, China; 2College of Electrical and Electronic Engineering, Nanyang Technological University, Singapore 639798, Singapore

**Keywords:** cognitive radio network, energy harvesting, physical layer security, deep reinforcement learning

## Abstract

The paper studies the secrecy communication threatened by a single eavesdropper in Energy Harvesting (EH)-based cognitive radio networks, where both the Secure User (SU) and the jammer harvest, store, and utilize RF energy from the Primary Transmitter (PT). Our main goal is to optimize the time slots for energy harvesting and wireless communication for both the secure user as well as the jammer to maximize the long-term performance of secrecy communication. A multi-agent Deep Reinforcement Learning (DRL) method is proposed for solving the optimization of resource allocation and performance. Specifically, each sub-channel from the Secure Transmitter (ST) to the Secure Receiver (SR) link, along with the jammer to the eavesdropper link, is regarded as an agent, which is responsible for exploring optimal power allocation strategy while a time allocation network is established to obtain optimal EH time allocation strategy. Every agent dynamically interacts with the wireless communication environment. Simulation results demonstrate that the proposed DRL-based resource allocation method outperforms the existing schemes in terms of secrecy rate, convergence speed, and the average number of transition steps.

## 1. Introduction

Cognitive Radio (CR) is regarded as a potential solution for spectrum resource scarcity as a result of the extensive use of wireless technology, the growing demand for high-speed data transmissions and the traditional static spectrum strategies [1]. In Cognitive Radio Networks (CRNs), cognitive users are able to utilize licensed spectrum resources by underlay, overlay or interweave modes. In the underlay mode, the cognitive users are allowed to access the licensed spectrum occupied by the Primary Users (PUs) only when the interference temperature to PUs is lower than a desired level [2].

However, battery-limited devices in CRNs will bring great inconvenience [3], e.g., for the key elements embedded inside human bodies or wireless sensors working under extreme environments, replacing or recharging their batteries is not accessible. Luckily, Energy Harvesting (EH) technique has appeared as an exciting solution to this issue. EH refers to harvesting energy from the environments (e.g., via thermal, wind, solar, and wireless Radio Frequency (RF) energy sources) and then converting it into electric power energy for self-maintenance circuits [4]. Compared with natural energy supply, RF energy is capable of providing continuous, stable, and clean power to CRN terminals. Therefore, using RF energy to supply cognitive wireless networks is a significant technology for raising spectrum utilization and energy efficiency in CRNs.

Despite the aforementioned advantages, the CRN system is often subjected to illegal wiretapping as a result of characteristics of wireless channels [5]. In recent years, owing to rapid development in computational capacity, traditional cryptography encryption techniques are easily decoded by illegitimate users. Therefore, Physical Layer Security (PLS) approach has become an alternative technology for secure transmissions [6]. PLS technique aims to ensure secrecy performance, e.g., secrecy rate means a communication rate at which secrecy signals could be transmitted from a transmitter to an expected receiver [7]. In the information security theory, secrecy capacity refers to the maximum achievable secrecy rate. Once the secrecy capacity is worse than zero, the communications between the transmitter and the receiver are at risk, and eavesdroppers would be able to wiretap secrecy signals transmitted by the transmitter [8]. The method significantly improves the security of communications by diminishing the wiretapping capacity of eavesdroppers [7]. The broadcast characteristics of wireless channels also provide the opportunity to introduce interference into transmissions to reduce the wiretapping ability of an eavesdropper while enhancing the ability of both legitimate users to communicate securely [9]. To this end, Artificial Noise (AN) and Cooperative Jamming (CJ) have emerged as promising approaches for enhancing secrecy performance. The former realizes the process by mixing the AN signal, which acts as jamming signals, into the confidential information signals to interfere with eavesdroppers. In contrast, the latter realizes it by directly sending the jamming signals from the cooperative jammer to hinder the wiretapping channels and weaken the capabilities for decoding the wiretapped information [10]. If the legal receivers support full-duplex communications, it is technically feasible to transmit jamming signals to raise system performance benefits [11], and, furthermore, the CJ technology will be more positive and efficient once the eavesdroppers get closer to the legal receiver [12]. In addition, there are also beamforming and precoding secure transmission methods; however, the complexity of the Beamforming and Precoding schemes in the actual wireless communication system is critical to the operation of the system, and the extremely high computational complexity makes it difficult to apply it in practical systems. In the research of existing papers, CJ is one of the most important ways to achieve secure PLS transmission. In the CJ secure transmission scheme, the jammer can complete the design of jamming signals beamforming vector by using the statistical Channel State Information (CSI) of illegal channels, which is more suitable for actual wireless communication scenarios. Considering the above points, in this paper, we apply the CJ method to our proposed network. The research on physical layer security is usually divided into two cases: one is that the CSI of the eavesdropper is known, and the other is that the channel state information is not perfect. In most practical cases, the accurate location and CSI of the eavesdropper are unknown to the network. Our work considers the second case, which is a common assumption in the field of physical layer security.

## 2. Related Work

In the last few years, many explorations on combining the EH and CR techniques for the purpose of improving secrecy communication have been conducted. In [13], the authors investigate the communication security in an EH-based cooperative CRN, where the cognitive source and cognitive relays are capable of harvesting RF energy from the surrounding environment and derive the closed-form expressions of secrecy outage probability via the proposed two relay selection schemes. In [14], the authors investigate the security and efficiency of data transmission for overlay CRNs and then propose an optimal relay selection solution on the basis of two EH schemes for a balance between the secrecy performance and the efficiency of communication transmissions. Unlike the overlay spectrum access considered in [14], the authors in [15] study an underlay CRN which consists of a Secondary Base Station (SBS), a PU, and multiple energy-constrained secondary users. The secondary users first harvest RF energy signals emitted by the Primary Transmitter (PT) and then communicate with the SBS via the proposed user scheduling schemes. The authors in [16] study an Unmanned Aerial Vehicle (UAV)-aided Energy Harvesting CRN in which an EH-enabled UAV as a secondary user first harvests energy, performs spectrum sensing and then communicates with a ground destination. For maximizing the outage energy efficiency for the CRN, a resource allocation policy is utilized to solve the proposed optimization problem. Linear EH in [13,14,15] is an ideal acquisition model, while the non-linear EH model can better reflect the actual situation of EH. Different from the linear EH model in [13,14,15] and the random energy arrival model in [16], the authors in [17] consider a more realistic nonlinear EH model and jointly optimize the EH time-slot, channel selection, and transmission power by utilizing the 1-D searching algorithm. Numerical results show the secrecy performance acquired by the nonlinear EH model is equivalent to that acquired by the linear EH model. Similar to [17], the authors in [18] study a MIMO non-orthogonal multiple access CRN with a realistic non-linear EH model. An AN-aided beamforming design problem with the practical secrecy rate and EH causality constraint is explored, and then the semidefinite relaxation-based and cost function algorithms are proposed to solve the proposed problems.

The formulation of physical layer security enhancement in EH-CRN in prior works can be modeled as the optimization of resource allocation problems. However, many existing documents only focus on conventional resource management and allocation approaches which present poor efficiency and high computational complexity in seeking the optimal strategies. As a matter of fact, the objection function in the optimization problem often exhibits a non-convex property; therefore, it is difficult to be solved by traditional optimization theory. Therefore, a more efficient and intelligent solution for resource allocation is needed.

With the rapid growth of deep learning recently, Deep Reinforcement Learning (DRL) also has gradually been rising and developing. Up to now, it has been extensively applied to numerous sophisticated communication systems for providing high-quality services; for instance, the authors in [19] introduce the DRL method to the 5G communication systems for network slicing by dynamically allocating system resources for a wide range of services over a common underlying physical infrastructure. A lot of work has validated its superiority in improving the security of communications by achieving resource management and allocation in existing wireless communication networks. In [20], the communication security of a UAV-assisted relay selection CRN is studied by transmitting the AN signal to the eavesdropper, and then a Q-learning algorithm is proposed for a dynamic power allocation problem. Q-Learning algorithm is a typical model-free algorithm; the update of its Q table is different from the strategy followed when selecting actions; in other words, when updating the Q table, the maximum value of the next state is calculated, but the action corresponding to the maximum value does not depend on the current strategy. However, the Q-learning algorithm faces two main problems: (1) when the state space is large, Q-learning will consume a lot of time and space for searching and storage; (2) Q-learning has the problem of overestimation for state-action value. Considering the fatal shortcomings of the Q-learning algorithm, we further turn to the research of the DRL algorithm.

Unlike the research in [20] that only focuses on a traditional reinforcement learning algorithm, Ref. [21] proposes a Multi-Agent Deep Deterministic Policy Gradient (MADDPG) algorithm for maximizing the secrecy capacity by joint optimization of the UAVs’ trajectory, the transmission power, and the jamming power. The proposed MADDPG method is an extension of DDPG in multi-agent tasks. Its basic idea is centralized learning and decentralized execution. During training, the actor network is updated by a global critical network, while during testing, the actor network takes action. MADDPG has a distinct advantage in dealing with the continuous state and action spaces, compared with the traditional reinforcement learning method in [20]. However, the MADDPG algorithm is also susceptible to large state space, e.g., the large state space easily leads to an unstable convergence process. In [22], for maximizing the long-term achievable secrecy rate while minimizing energy consumption, a UAV-to-vehicle communication scenario with multiple eavesdroppers on the ground is modeled as a Markov Decision Process (MDP), and then solved by a Curiosity-driven Deep Q-network (C-DQN) algorithm. On the one hand, the C-DQN algorithm uses a deep convolutional neural network to approximate the state-action value function, so the model has better robustness; on the other hand, C-DQN also has the same overestimation problem as the Q-learning algorithm. Different from the communication scenario portrayed by [22], the authors in [23] consider a relay-assisted mobile system. For maximizing the average secrecy rate, several model-free reinforcement learning-based algorithms are developed for seeking optimal resource allocation strategies. Specifically, the UAV relay equipped with multiple antennas is modeled as an agent. The proposed algorithms can enable the agent to explore the characteristics of the environment through a large number of interactions, thus, derive the optimal UAV trajectory policy. In the above literature, although the Q-learning algorithm in [20] and the C-DQN algorithm in [22] can improve the system performance to some extent, there exists the overestimation of the state-action value. There is an urgent need for an improved DRL algorithm that can overcome the overestimation and improve system performance to the greatest extent.

## 3. Motivation and Contributions

The DRL methods show great advantages in dealing with various resource allocation problems in secure secrecy communication scenarios. However, to the best of our knowledge, few papers have studied secrecy communications in EH-CRN networks by using the DRL methods for joint scheduling of EH time slots and transmission power. Moreover, motivated by the prior works, we focus our study in this paper on the PLS enhancement in EH-CRN systems in the presence of a potential eavesdropper by combining CR, EH, and PLS techniques for the following main reasons. Firstly, the CR technique allows SUs to utilize the licensed spectrum of PU. Secondly, the EH technique is utilized to provide a sustainable energy supply for energy-constrained nodes, i.e., Secure Transmitter (ST) and jammer. More importantly, the throughput of the proposed network can be significantly improved through the EH technique, and the jammer also becomes more aggressive in defending against eavesdropping. Thirdly, the PLS technique is used to achieve secure communication of the SU to transmit secrecy information. Although the authors in [20] consider the combination of CR, EH, and PLS techniques, the traditional reinforcement learning algorithm in [20] can only deal with the discrete state space, and it appears to be too idealistic to reflect the real situation, so it is necessary to consider use the DRL method to study a more reasonable and practical continuous state space.

Moreover, Refs. [21,22,23] do not consider the EH technique for their systems. Furthermore, the proposed DRL-based resource allocation method for PLS enhancement basically differs from these existing papers [20,21,22,23] in the followings: A multi-agent DRL framework for the proposed EH-CRN is modeled; The classical DRL algorithm is combined with the Long Short-Term Memory Network (LSTM) for acquiring more performance improvements; The proposed algorithm is equipped with favorable stability and fast convergence speed.

The main contributions of this paper are listed in the following.
We consider an EH-CRN, where the communication security of a legitimate user is under threat, and a cooperative jammer is deployed to improve the system’s secrecy performance. The main goal is to enhance the physical layer security by achieving the optimal resource allocation for our proposed network.For tackling the joint EH time slot and power allocation issue, we propose a multi-agent DRL-based resource allocation framework for our proposed network. Specifically, we model two types of agents, and each agent interacts with the dynamic environment through the state, action, and reward mechanisms.To improve the classical DRL algorithm performance, we propose a new DRL network architecture, where the LSTM architecture is incorporated into the Dueling Double Deep Q-Network (D3QN) algorithm for overcoming the negative effects of the dynamic characteristics of the network caused by the time-varying channels and the random noise. Moreover, the construction of the loss function in the proposed method is different from the previously mentioned algorithms, and, thus, it can well avoid the overestimation of the state-action value and make the training process more stable.Based on presented experimental results, the proposed scheme is efficient in improving the long-term achievable secrecy rate with small training episodes overheads.

## 4. Paper Organization

The remainder of the paper is organized as follows. Section 5 presents an EH-based CRN system model and a related optimization problem. Section 6 proposes a multi-agent DRL framework to obtain a solution to this joint EH time and power allocation problem. Section 7 presents simulation results to evaluate the system performance via our proposed method as compared to benchmark schemes. Finally, Section 8 concludes this paper.

## 5. System Model and Problem Formulation

### 5.1. System Model

As shown in Figure 1, we examine a secrecy communication in a CRN with a PU which consists of a PT and a Primary Receiver (PR), a SU which consists of an ST and a Secure Receiver (SR), a cooperative jammer which enables jamming signals, a potential eavesdropper who attempts to eavesdrop the secrecy data transmitted by the ST. The SU is entitled to utilize the licensed spectrum to the PU using underlay mode. The ST and the jammer are equipped with energy harvesters and batteries, respectively. The energy harvester can collect RF energy signals from the PT, and store this energy in the battery. We adopt a block-based quasi-static model; that is, the wireless CSI remains unchanged over each transmission block but may vary from one block to another [7]. The information link refers to the channels of PT-PR and ST-SR, while the wiretapping link refers to the channels of ST-Eavesdropper. The EH link refers to the channels of PT-ST and PT-Jammer. The interfering link refers to the channels of PT-SR and ST-PR. The jamming link refers to the channels of Jammer-Eavesdropper. All nodes carry only one antenna.

To destroy the eavesdropping capability, the SU and the jammer have perfect knowledge of the CSIs for the wiretapping link, and the jamming link over each transmission block [6,22,24]. It is assumed that the jamming signal from the jammer can be eliminated by the SR but cannot be removed at the eavesdropper [25,26]. This can be realized by the following method: A large number of stochastic sequences (jamming signals) with Gaussian distribution are pre-stored at the jammer, and their indices act as the keys. The jammer stochastically picks a sequence (jamming signal) and transmits its key to the SR. The key can be transmitted in a secret way via channel independence and reciprocity. As the stochastic sequence is only known at the SR, any eavesdropper is unable to extract the stochastic sequence. We suppose that the ST will be given an indicator signal which signifies whether the received Quality of Service (QoS) by the PR is satisfied [27].

A transmission link is made of multiple subcarriers, and let M≜{1,2,…,M} as the set of these subcarriers. We denote $G_{\mathrm{PP}}$≜$\left\{g_{\mathrm{PP}}^{m}|m\in M\right\}$, $G_{\mathrm{SS}}$≜$\left\{g_{\mathrm{SS}}^{m}|m\in M\right\}$, $I_{\mathrm{SP}}$≜$\left\{i_{\mathrm{SP}}^{m}|m\in M\right\}$, $I_{\mathrm{PS}}$≜$\left\{i_{\mathrm{PS}}^{m}|m\in M\right\}$, $W_{\mathrm{SE}}$≜$\left\{w_{\mathrm{SE}}^{m}|m\in M\right\}$, $H_{\mathrm{PS}}$≜$\left\{h_{\mathrm{PS}}^{m}|m\in M\right\}$, $H_{\mathrm{PJ}}$≜$\left\{h_{\mathrm{PJ}}^{m}|m\in M\right\}$, and $J_{\mathrm{JE}}$≜$\left\{j_{\mathrm{JE}}^{m}|m\in M\right\}$ as the channel gain coefficients sets from the PT-PR, the ST-SR, the ST-PR, the PT-SR, the ST-Eavesdropper, the PT-ST, the PT-Jammer, and the Jammer-Eavesdropper links. Different fading subchannels in each link are independent and identically distributed Rayleigh distributed random variables with mean zero and variances one. Over the *m*-th subcarrier, let $p_{\mathrm{PT}}^{m}$, $p_{\mathrm{ST}}^{m}$ and $p_{J}^{m}$ indicate the RF power of the PT, the transmission powers of the ST and the jammer, respectively. Denote, respectively, the RF energy signal, the secrecy information signal from the ST, and the jamming signal from the jammer as $S_{\mathrm{PT}}^{m}$, $S_{\mathrm{ST}}^{m}$ and $S_{J}^{m}$, which are independent cyclic symmetric complex Gaussian random variables with mean zero and different variances $E(|S_{\mathrm{PT}}^{m}{|}^{2})=p_{\mathrm{PT}}^{m}$, $E(|S_{\mathrm{ST}}^{m}{|}^{2})=p_{\mathrm{ST}}^{m}$ and $E(|S_{J}^{m}{|}^{2})=p_{J}^{m}$. E(·)=∫(·)f(x)dx is the statistic expectation.

As shown in Figure 2, we consider a two-phase transmission scheme for both the ST and the jammer by dividing each transmission block into two time slots. In the first time slot $T_{1}$, namely the EH phase, the PT broadcasts wireless RF energy signals to its receiver. In the meantime, the ST and the jammer harvest and store RF energy in their batteries, respectively. In the second time slot $T_{2}$, namely the data transmission phase, the PU instantaneously updates its transmission power based on a power control strategy but is unknown to the SU [27], and transmits the public broadcasting signals to the PR, because the public broadcasting signal is meaningless to eavesdroppers, this paper only considers that the signal received by the eavesdroppers only includes the secrecy signal of SU. The ST transmits secrecy information to the SR while the eavesdropper begins to wiretap the secrecy information. To ensure secure communication, the jammer instantaneously sends jamming signals to intercept its wiretapping. The length of each transmission block is *T* and includes an EH duration and a data transmission duration. We denote $\alpha_{1}^{t}$ and $\beta_{1}^{t}$ as portions of two time slots of the ST over a transmission block, $\alpha_{2}^{t}$, and $\beta_{2}^{t}$ for the jammer. Therefore, at the ST and the jammer, we have
(1)$$\begin{array}{c} \alpha_{1}^{t}+\beta_{1}^{t}=1,0\leq \alpha_{1}^{t}\leq 1,0\leq \beta_{1}^{t}\leq 1 \end{array}$$
and
(2)$$\begin{array}{c} \alpha_{2}^{t}+\beta_{2}^{t}=1,0\leq \alpha_{2}^{t}\leq 1,0\leq \beta_{2}^{t}\leq 1. \end{array}$$

The RF powers received by the ST and the jammer over the *m*-th subcarrier are given by
(3)$$\begin{array}{c} p_{\mathrm{ST},\mathrm{received}}^{m}=\alpha_{1}^{t}\eta_{1}p_{\mathrm{PT}}^{m}{\left|h_{\mathrm{PS}}^{m}\right|}^{2},\alpha_{1}^{t}<\frac{T_{1}}{T} \end{array}$$
and
(4)$$\begin{array}{c} p_{J,\mathrm{received}}^{m}=\alpha_{2}^{t}\eta_{2}p_{\mathrm{PT}}^{m}{\left|h_{\mathrm{PJ}}^{m}\right|}^{2},\alpha_{2}^{t}<\frac{T_{1}}{T}, \end{array}$$
respectively. Here, $\eta_{1}$ and $\eta_{2}$ represent the EH efficiency at the ST and the jammer, respectively. According to the traditional non-linear EH model, the harvested energy of the ST and the jammer are respectively expressed as
(5)$$\begin{array}{c} E_{\mathrm{NL},l}^{m}=\frac{\Gamma_{l}^{m}-A_{l}\Psi_{l}}{1-\Psi_{l}}, \end{array}$$
(5a)$$\begin{array}{c} \Psi_{l}=\frac{1}{1+\exp(a_{l}b_{l})} \end{array}$$
and
(5b)$$\begin{array}{c} \Gamma_{l}^{m}=\frac{A_{l}}{1+\exp(-a_{l}\left(p_{l,\mathrm{received}}^{m}-b_{l}\right))}, \end{array}$$
where $\Gamma_{l}^{m},l=\left\{\mathrm{ST},J\right\}$ is a traditional logic function related to the received RF power $p_{l,\mathrm{received}}^{m}$. Parameters $a_{l}$ and $b_{l}$ are related to the specification of the specific EH circuits. $A_{l}$ represents the maximum EH power received by the energy receiver when the EH process reaches a saturation [17,28]. Owing to the fact that an ideal linear EH model is unable to reflect the practical EH situation. Consequently, this paper considers a non-linear EH model for the proposed network in this paper.

Over the *m*-th subcarrier, the received signals by the PR, the SR, and the eavesdropper are, respectively, denoted by
(6)$$\begin{array}{c} y_{\mathrm{PR}}^{m}=g_{\mathrm{PP}}^{m}S_{\mathrm{PT}}^{m}+i_{\mathrm{SP}}^{m}S_{\mathrm{ST}}^{m}+n_{\mathrm{PR}}^{m}, \end{array}$$
(7)$$\begin{array}{c} y_{\mathrm{SR}}^{m}=g_{\mathrm{SS}}^{m}S_{\mathrm{ST}}^{m}+i_{\mathrm{PS}}^{m}S_{\mathrm{PT}}^{m}+n_{\mathrm{SR}}^{m} \end{array}$$
and
(8)$$\begin{array}{c} y_{E}^{m}=w_{\mathrm{SE}}^{m}S_{\mathrm{ST}}^{m}+j_{\mathrm{JE}}^{m}S_{J}^{m}+n_{E}^{m}, \end{array}$$
where $n_{\mathrm{PR}}^{m}$, $n_{\mathrm{SR}}^{m}$, and $n_{E}^{m}$ denote Gaussian noise signals received by the PR, the SR, and the eavesdropper with mean zero and variances $E(|n_{\mathrm{PR}}^{m}{|}^{2})=E(|n_{\mathrm{SR}}^{m}{|}^{2})=E(|n_{E}^{m}{|}^{2})=1$, respectively.

At the ST and the jammer, we have maximum transmission power constraints
(9)$$\begin{array}{c} 0\leq \sum_{m\in M}p_{\mathrm{ST}}^{m}\leq p_{\mathrm{ST},\max} \end{array}$$
and
(10)$$\begin{array}{c} 0\leq \sum_{m\in M}p_{J}^{m}\leq p_{J,\max}, \end{array}$$
where $p_{\mathrm{ST},\max}$ and $p_{J,\max}$ denote maximum tolerable transmission powers at the ST and the jammer, respectively.

The QoS constraints at the receivers, that the received Signal-to-Interference-plus-Noise-Ratio (SINR) by the SR and the PR are, respectively, no lower than their minimum levels $\lambda_{1}$, $\lambda_{2}$, can be represented by
(11)$$\begin{array}{c} {\mathrm{SINR}}_{\mathrm{SR}}=\sum_{m\in M}\frac{p_{\mathrm{ST}}^{m}{\left|g_{\mathrm{SS}}^{m}\right|}^{2}}{p_{\mathrm{PT}}^{m}{\left|i_{\mathrm{PS}}^{m}\right|}^{2}+1}\geq \lambda_{1} \end{array}$$
and
(12)$$\begin{array}{c} {\mathrm{SINR}}_{\mathrm{PR}}=\sum_{m\in M}\frac{p_{\mathrm{PT}}^{m}{\left|g_{\mathrm{PP}}^{m}\right|}^{2}}{p_{\mathrm{ST}}^{m}{\left|i_{\mathrm{SP}}^{m}\right|}^{2}+1}\geq \lambda_{2}. \end{array}$$

The energy causal constraint for the ST and the jammer that the consumed energy for transmitting or jamming in the second time slot cannot exceed the currently available battery capacity is given as
(13)$$\begin{array}{c} E_{\mathrm{NL},\mathrm{ST}}^{t}+B_{\mathrm{ST}}^{t-1}-\beta_{1}^{t}T\sum_{m\in M}p_{\mathrm{ST}}^{m}\geq 0, \end{array}$$
(14)$$\begin{array}{c} E_{\mathrm{NL},J}^{t}+B_{J}^{t-1}-\beta_{2}^{t}T\sum_{m\in M}p_{J}^{m}\geq 0, \end{array}$$
(15)$$\begin{array}{c} 0\leq B_{\mathrm{ST}}^{t}\leq B_{\mathrm{ST},\max}, \end{array}$$
(16)$$\begin{array}{c} 0\leq B_{J}^{t}\leq B_{J,\max}, \end{array}$$
where $B_{\mathrm{ST},\max}$ and $B_{J,\max}$ are the maximum battery capacity, $B_{\mathrm{ST}}^{t-1}$ and $B_{J}^{t-1}$ stand for the residual battery capacity of the ST and the jammer at the transmission block t−1, respectively.

In conclusion, the secrecy rate $R_{\sec}\left[t\right]$ at each transmission block *t* is defined by
(17)$$\begin{array}{c} R_{\sec}\left[t\right]=\sum_{m\in M}{\left[R_{\mathrm{SR}}^{\left(m\right)}\left[t\right]-R_{E}^{\left(m\right)}\left[t\right]\right]}^{+}, \end{array}$$
where ${\left[\cdot \right]}^{+}≜\max\left(0,\cdot \right)$, the achievable rate $R_{\mathrm{SR}}^{\left(m\right)}\left[t\right]$ on the ST-SR link and the wiretapping rate $R_{E}^{\left(m\right)}\left[t\right]$ on the ST-Eavesdropper and Jammer-Eavesdropper links over each transmission block *t* and *m*-th subcarrier are, respectively, expressed as
(18)$$\begin{array}{c} R_{\mathrm{SR}}^{\left(m\right)}\left[t\right]=\beta_{1}^{t}\log_{2}\left(1+\frac{p_{\mathrm{ST}}^{m}{\left|g_{\mathrm{SS}}^{m}\right|}^{2}}{p_{\mathrm{PT}}^{m}{\left|i_{\mathrm{PS}}^{m}\right|}^{2}+1}\right) \end{array}$$
and
(19)$$R_{E}^{\left(m\right)}\left[t\right]=\left\{\begin{array}{ccc} R_{m}^{\left(1\right)}\left[t\right], & \; & \alpha_{1}^{t}\geq \alpha_{2}^{t}\\ R_{m}^{\left(2\right)}\left[t\right], & \; & \mathrm{otherwise} \end{array}\right.,$$
where
(19a)$$\begin{array}{c} R_{m}^{\left(1\right)}\left[t\right]=\left(1-\alpha_{1}^{t}\right)\log_{2}\left(1+\frac{p_{\mathrm{ST}}^{m}{\left|w_{\mathrm{SE}}^{m}\right|}^{2}}{p_{J}^{m}{\left|j_{\mathrm{JE}}^{m}\right|}^{2}+1}\right) \end{array}$$
and
(19b)$$\begin{array}{cc} R_{m}^{\left(2\right)}\left[t\right]= & \left(\alpha_{2}^{t}-\alpha_{1}^{t}\right)\log_{2}\left(1+p_{\mathrm{ST}}^{m}{\left|w_{\mathrm{SE}}^{m}\right|}^{2}\right)+\left(1-\alpha_{2}^{t}\right)\log_{2}\left(1+\frac{p_{\mathrm{ST}}^{m}{\left|w_{\mathrm{SE}}^{m}\right|}^{2}}{p_{J}^{m}{\left|j_{\mathrm{JE}}^{m}\right|}^{2}+1}\right) \end{array}$$

### 5.2. Problem Formulation

In general, there exists a tradeoff between EH and WIT, e.g., for the jammer, a longer EH duration can harvest more energy signals to increase the jamming power for fighting against illegal eavesdropping; but, for the ST, it can reduce the EH duration to acquire more available WIT duration for delivering confidential information at next transmission block. Our goal is to seek an optimal joint EH time coefficient and transmission power allocation strategy over each transmission block for maximizing the long-term achievable secrecy rate while maintaining other constraint requirements. The comprehensive problem can be formulated as
(20)$$\begin{array}{cc} \; & \max_{\alpha_{1}^{t},\alpha_{2}^{t},p_{\mathrm{ST}}^{m},p_{J}^{m}}\;\;E\left\{\sum_{t=1}^{\infty }R_{\sec}\left[t\right]\right\}\;\\ \; & \begin{array}{c} s.t.(1),(2),(3),(4),(9),(10),(11),(12),(13),(14),(15),(16). \end{array} \end{array}$$

It is observed that the proposed problem is non-convex as the objective function is non-concave, and, thus, an effective solution is needed.

## 6. Deep Reinforcement Learning for Joint Time and Power Allocation

### 6.1. DRL Framework

To solve the proposed problem, this paper proposes a multi-agent DRL framework for it, as shown in Figure 3. The DRL framework is composed of an environment and multiple agents. Each sub-channel from the ST-SR and the Jammer-Eavesdropper links is regarded as an agent which aims to explore optimal transmission power allocation strategy. Without loss of generality, a time allocation network as a “time” agent is established to obtain optimal EH time coefficients $\left\{\alpha_{1}^{t},\alpha_{2}^{t}\right\}$, respectively. Let $K≜\left\{0,1,2,\ldots ,2M\right\}$ as the set of 2M+1 agents. The others in this framework are regarded as the environment. The agents are collectively interacting with the dynamic environment to acquire a large number of learning experiences for seeking the optimal resource allocation policy. Such an interactive process can be modeled as an MDP (S,A,R,P,γ), where S is the state space, A is the action space, R is the reward function, P is a state transition probability and γ∈[0,1) is a reward discount factor. A transmission block is regarded as a time step. At each time step *t*, once given a current environment state $S_{t}$, each agent k∈K observes a local observation $Z_{t}^{k}=O\left(S_{t},k\right)$, and then takes an action $a_{t}^{k}$. As a result, the agent receives a reward $r_{t}^{k}$ from the environment, and the current state $S_{t}$ instantaneously transfers to the following state $S_{t+1}$ according to the probability P.

### 6.2. Observation Space

To activate agents to learn an effective strategy $\pi (A_{t}|S_{t})$, the current environment state $S_{t}$ must reflect the environment characteristics as much as possible. The jointly instantaneous CSIs from different transmission links can be expressed as
(21)$$\begin{array}{cc} G_{t}= & {\left\{\{g_{\mathrm{SS}}^{m}\right\}}_{m\in M},{\left\{i_{\mathrm{PS}}^{m}\right\}}_{m\in M},{\left\{h_{\mathrm{PS}}^{m}\right\}}_{m\in M},{\left\{g_{\mathrm{PP}}^{m}\right\}}_{m\in M},\\ \; & {\left\{i_{\mathrm{SP}}^{m}\right\}}_{m\in M},{\left\{h_{\mathrm{PJ}}^{m}\right\}}_{m\in M},{\left\{w_{\mathrm{SE}}^{m}\right\}}_{m\in M},{\left\{j_{\mathrm{JE}}^{m}\right\}}_{m\in M}\}, \end{array}$$

At the transmission block t−1, we denote
(22)$$\begin{array}{c} I_{t-1}=\left\{\sum_{m\in M}p_{\mathrm{PT}}^{m}{\left|i_{\mathrm{PS}}^{m}\right|}^{2},\sum_{m\in M}p_{\mathrm{ST}}^{m}{\left|i_{\mathrm{SP}}^{m}\right|}^{2},\sum_{m\in M}p_{J}^{m}{\left|j_{\mathrm{JE}}^{m}\right|}^{2}\right\}, \end{array}$$
(23)$$\begin{array}{c} {\mathrm{SINR}}_{t-1}=\left\{{\mathrm{SIN}R}_{\mathrm{PR}},{\mathrm{SIN}R}_{\mathrm{SR}}\right\}, \end{array}$$
and
(24)$$\begin{array}{c} B_{t-1}=\left\{B_{\mathrm{ST}}^{t-1},B_{J}^{t-1}\right\} \end{array}$$
as joint interference powers, the joint SINRs, and the joint residual battery capacity, respectively. The instantaneous CSIs are included to reflect the current channel state. The joint interference power $I_{t-1}$ is related to agents’ transmission powers, which have a straight impact on the environment. For instance, the transmission power of the ST may cause strong interference to the main link, under which the SU may be inhibited from accessing the licensed spectrum to the PU and, thus, straightly leads to a temporary secrecy rate reduction. The joint SINRs at the previous transmission block represents the received QoS by the PR and the SR, and it facilitates the improvement of the power strategy of SU. As current battery capacity $B_{\mathrm{ST}}^{t}$ is related to the residual capacity $B_{\mathrm{ST}}^{t-1}$, the transmission power at the ST will be influenced by $B_{\mathrm{ST}}^{t-1}$. With further analysis of the state mechanism, the previous reward $r_{t-1}$ can be acted as feedback to activate agents, and, thus, the reward $r_{t-1}$ is included in the observation $Z_{t}^{k}.$

In conclusion, the observation function $Z_{t}^{k}$ of each agent *k* at the time step *t* is given as
(25)$$\begin{array}{c} Z_{t}^{k}=\left\{G_{t},I_{t-1},{\mathrm{SINR}}_{t-1},B_{t-1},r_{t-1}\right\}. \end{array}$$

### 6.3. Action Space

We denote $a_{t}^{k}$ as the action of the agent *k* at the time step *t*. The joint action of the agents is formulated as
(26)$$\begin{array}{c} A_{t}=\left\{a_{t}^{0},a_{t}^{1},\ldots ,a_{t}^{M},\ldots ,a_{t}^{2M}\right\}. \end{array}$$


The action of the “time” agent is set as the EH time coefficients $C_{t}=\left\{\alpha_{1}^{t},\alpha_{2}^{t}\right\}$ and the actions of other agents are set as their transmission powers. Therefore, the joint action of all agents is expressed as
(27)$$\begin{array}{c} A_{t}=\left\{\left\{\alpha_{1}^{t},\alpha_{2}^{t}\right\},p_{\mathrm{ST}}^{1},\ldots ,p_{\mathrm{ST}}^{M},p_{J}^{1},\ldots ,p_{J}^{M}\right\}. \end{array}$$

To alleviate the effects during the process of learning, optional EH time coefficients $\alpha_{1}^{t}$ and $\alpha_{2}^{t}$ are discretized as $L_{1}$ time levels, i.e., $c_{1},c_{2},\ldots ,c_{L_{1}}$; optional transmission powers are discretized as $L_{2}$ power levels, i.e., $p_{1},p_{2},\ldots ,p_{L_{2}}.$

### 6.4. Reward Design

We convert some constraint requirements in the proposed problem into a part of the reward. A reward consists of an instantaneous secrecy rate $R_{\sec}\left[t\right]$ at current transmission block *t*, the joint SINRs at previous transmission block t−1, and the battery capacity $B_{\mathrm{ST}}^{t}$, $B_{J}^{t}$. In conclusion, the reward for each agent *k* is expressed as
(28)$$\begin{array}{c} r_{t}=\eta_{1}R_{\sec}\left[t\right]+\eta_{2}R_{\mathrm{SINR}}\left[t\right]+\eta_{3}R_{\mathrm{Bac}}\left[t\right] \end{array}$$
where
(28a)$$\begin{array}{cc} R_{\mathrm{SINR}}\left[t\right]= & \left(\sum_{m\in M}\frac{p_{\mathrm{ST}}^{m}{\left|g_{\mathrm{SS}}^{m}\right|}^{2}}{p_{\mathrm{PT}}^{m}{\left|i_{\mathrm{PS}}^{m}\right|}^{2}+1}-\lambda_{1}\right)+\left(\sum_{m\in M}\frac{p_{\mathrm{PT}}^{m}{\left|g_{\mathrm{PP}}^{m}\right|}^{2}}{p_{\mathrm{ST}}^{m}{\left|i_{\mathrm{SP}}^{m}\right|}^{2}+1}-\lambda_{2}\right) \end{array},$$
(28b)$$\begin{array}{c} R_{\mathrm{Bac}}\left[t\right]=\left(B_{\mathrm{ST}}^{t-1}-0.1B_{\mathrm{ST},\max}\right)+\left(B_{J}^{t-1}-0.1B_{J,\max}\right) \end{array},$$
(28c)$$\begin{array}{c} \eta_{1}+\eta_{2}+\eta_{3}=1,0\leq \eta_{1},\eta_{2},\eta_{3}\leq 1, \end{array}$$
and 0.1 in (28b) is a critical threshold of the battery capacity.


In this reward $r_{t}$, the first entry $R_{\sec}\left[t\right]$ is a performance-oriented part, which directs an agent’s learning direction. The second entry $R_{\mathrm{SINR}}\left[t\right]$ is related to the QoS of the SR and the PR. There is a balance between an instantaneous secrecy rate and available battery capacity; therefore, the third entry $R_{\mathrm{Bac}}\left[t\right]$ is necessary to be added into $r_{t}$. Considering different impacts on system performance, each entry is endowed with a positive weight that ranges from zero to one.

At each time step *t*, the long-term expected return $R_{t}$ for an agent *k* is defined as
(29)$$\begin{array}{c} R_{t}=E\left[\sum_{l=0}^{\infty }w^{l}r_{t+1}\right], \end{array}$$
where w∈[0,1] is a discount factor. In DRL, the main goal is to maximize the return $R_{t}$ by seeking an optimal strategy π.


### 6.5. LSTM-D3QN Algorithm

In our proposed system, the dynamic characteristics are primarily presented in time-varying channels, and the received random noise by the receivers. To overcome this issue, we resort to a combination of a classical DRL algorithm and the LSTM network, namely LSTM-D3QN, which is used to capture the temporal variation regularity of our proposed system. The LSTM-D3QN network architecture is presented in Figure 4. Each time step of the proposed algorithm is divided into two phases, i.e., the training phase and the implementation phase.
(1)Implementation phase

At the beginning of each episode, the environment state is randomly initialized. At each time step *t*, each agent *k* takes an action
(30)$$\begin{array}{c} a_{t}^{k}=\left\{\begin{array}{cc} \mathrm{random}(A) & \;0\leq p<\epsilon \\ {\mathrm{argmax}}_{a_{t}^{k}\in A}\left(Q\right) & \;\epsilon \leq p\leq 1 \end{array}\right. \end{array}$$
based on the ϵ-greedy policy, i.e., the optimal action $a_{t}^{k}$ is selected from the action space A with a probability 1−ϵ according to the maximal estimated action-value function $Q(Z_{t}^{k},a_{t}^{k})$ while a random action is derived with a probability ϵ. The collected transition $\left(Z_{t}^{k},a_{t}^{k},Z_{t+1}^{k},r_{t}\right)$ by the agent *k* is stored into the prioritized experience replay buffer $D_{k}$ when the environment has evolved from the current state $S_{t}$ to the next state $S_{t+1}.$
(2)Training Phase

Each agent *k* is a DRL algorithm model and, thus, has an LSTM-D3QN network architecture with a decision Q-network *Q* and a target Q-network $\hat{Q}$, which are initiated by parameters $\theta_{k}$ and ${\hat{\theta }}_{k}$, respectively. The action-value function of the decision Q-network is expressed as
(31)$$\begin{array}{c} Q(S_{t},a_{t}^{k};\theta_{1},\theta_{2})=V\left(S_{t};\theta_{1}\right)+\left(A\left(S_{t},a_{t}^{k};\theta_{2}\right)-\frac{1}{\left|A\right|}\sum_{a^{*}\in \left|A\right|}A\left(S_{t},a^{*};\theta_{2}\right)\right), \end{array}$$
where *V* is a state-value function with a parameter $\theta_{1}$ and *A* is an advantage function with a parameter $\theta_{2}$. The relationship between the value of taking an action *a* in current state *s* and the value of taking an action $a^{\prime }$ in the next state $s^{\prime }$ is described by the Bellman expectation equation
(32)$$\begin{array}{c} Q\left(s,a\right)=E_{\pi }[r_{t+1}+\gamma Q\left(S_{t+1}=s^{\prime },A_{t+1}=a^{\prime }\right)|S_{t}=s,A_{t}=a]. \end{array}$$

The structure of the target Q-network $\hat{Q}$ is the same as that of the decision Q-network *Q*. During the prioritized experience replay, the agent samples a mini-batch of transitions ${\left\{\left(s,a,s^{\prime },r\right)\right\}}_{i=1}^{K}$ from the replay buffer $D_{k}$ for updating the decision Q-network. The prioritized experience replay mechanism can accelerate the learning process by endowing each transition with different priorities. We define the TD-error for the replay buffer as
(33)$$\begin{array}{c} e=r_{t}+\gamma \hat{Q}\left(S^{\prime },a^{\prime };{\hat{\theta }}_{k}\right)-Q\left(S_{t},a_{t}^{k};\theta_{k}\right), \end{array}$$
where $S^{\prime },a^{\prime },S_{t},a_{t}^{k}\in D_{k}$. It will be more easily selected from $D_{k}$ as a training sample if a transition ($S_{t},a_{t}^{k},S_{t+1},r_{t}$) has a bigger absolute value |e|. The loss function of the decision Q-network *Q* is defined as a sum-squared error, that is
(34)$$\begin{array}{c} L\left(\theta_{k}\right)=\sum_{\left(S_{t},a_{t}^{k}\right)\in D_{k}}{\left(y-Q(S_{t},a_{t}^{k};\theta_{k})\right)}^{2}, \end{array}$$
where
(34a)$$\begin{array}{c} y=r_{t}+\gamma \hat{Q}\left(S_{t+1},a^{\prime };{\hat{\theta }}_{k}\right), \end{array}$$
(34b)$$\begin{array}{c} a^{\prime }=\underset{a^{\prime }\in A}{\mathrm{argmax}}Q\left(S_{t+1},a^{\prime };\theta_{k}\right). \end{array}$$

We apply the Adam optimizer with a learning rate δ to minimize the loss $L(\theta_{k})$ for updating the decision Q-network. For the target Q-network, the parameter ${\hat{\theta }}_{k}$ will be renewed once every $N_{\hat{Q}}$ time steps per episode by assigning current parameter $\theta_{k}$ to ${\hat{\theta }}_{k}$. The specific training procedure is described in the Algorithm 1.
**Algorithm 1** LSTM-D3QN with prioritized experience replay1:Start environment simulator and generate the network topology2:**Initialize** the channel gain of each link3:**Initialize** neural networks parameters randomly4:**Initialize** the capacity for each replay buffer $D_{k}$5:**for** each episode $e\in \{0,1,2,\ldots ,E_{\max}-1\}$ **do**6:     Update the locations of all nodes and the channel gains7:     Select randomly a joint EH time coefficient $C_{t}$8:     **Initialize** randomly transmission powers9:         **for** each step $t\in \{0,1,2,\ldots ,L_{\max}-1\}$ **do**10:          **for** each agent *k* **do**11:             Observe an observation $Z_{t}^{k}=O\left(S_{t},k\right)$ from             the current environment12:             Choose an action $a_{t}^{k}$ according to the ϵ-greedy             policy13:         **end for**14:         Update the channel gains15:         **for** each agent *k* **do**16:             Observe the next observation $Z_{t+1}^{k}$ and receive             the reward $r_{t}$ and then store the             transition $\left(Z_{t}^{k},a_{t}^{k},Z_{t+1}^{k},r_{t}\right)$ into $D_{k}$17:         **end for**18:      **end for**19:      **for** each agent *k* **do**20:         Sample a mini-batch of transitions from the replay         buffer $D_{k}$ and then update the decision Q-network21:         Renewing the parameter ${\hat{\theta }}_{k}$ of target Q-network         every $N_{\hat{Q}}$ time steps by assigning the current         parameter $\theta_{k}$ to ${\hat{\theta }}_{k}$22:      **end for**23:**end for**

### 6.6. Computational Complexity Analysis

The computational complexity of our proposed algorithm is mainly determined by the multiplications times in terms of the networks *Q* and $\hat{Q}$ [29]. We calculate the computational complexity of the implementation and training phases at each time step, respectively.

In the implementation phase of each time step, for an input *s* of the environment state, the network *Q* calculates out its corresponding output. On the basis of the connection and calculation theory about deep neural networks, the computational complexity of this process from input to output can be calculated as O(Ω), where $\Omega ≜f_{1}\left(I_{1}+W_{1}W_{2}\right)+f_{2}W_{2}+\sum_{i=2}^{3}f_{i}f_{i+1}+f_{5}\left(f_{2}+f_{6}\right)+O_{1}\left(f_{6}+1\right)$, $f_{l}$ is the number of neurons in the *l*-th (l=1,2,…,6) full connected layer, $W_{1}$ is the number of LSTM units, $W_{2}$ is the number of neurons in an LSTM unit, $I_{1}$ is the dimension of the input environment state, and $O_{1}$ is the number of neurons of the output layer which is equal to the size of the action space.

In the training phase of each time step, a minibatch of transition tuples ${\left\{\left(s,a,s^{\prime },r\right)\right\}}_{i=1}^{K}$ are sampled to update the network *Q*. Each episode contains *L* time steps, and the number of the agents is 2M+1. As the target network $\hat{Q}$ is updated every $N_{\hat{Q}}$ time steps, the computational complexity for 2M+1 agents in networks *Q* and $\hat{Q}$ during the training process of each episode is $O(\left(1+1/N_{\hat{Q}}\right)\left(2M+1\right)KL\Omega )$. During the prioritized experience replay, each state transition tuple $\left(s,a,s^{\prime },r\right)$ stored in buffer $D_{k}$ are sorted by the priority, and the corresponding computational complexity is $O(\sum_{k=1}^{2M+1}|D_{k}|\log_{2}\left(\right|D_{k}\left|)\right)$, where $|D_{k}|$ represents the capacity of buffer $D_{k}$. The total computational complexity for the whole training is calculated as $O(\left(1+1/N_{\hat{Q}}\right)\left(2M+1\right)KL\Omega +KL\sum_{k=1}^{2M+1}|D_{k}|\log_{2}\left(\right|D_{k}\left|)\right)$.

### 6.7. Convergence Analysis

The convergence of the Double Q-learning algorithm is the prerequisite for the convergence of the proposed algorithm. The Double Q-learning algorithm includes two functions: $Q_{A}$ and $Q_{B}$.

**Theorem** **1.**
*If Double Q-learning based DRL algorithm meets these following conditions: (C1): a large number of state transition tuples ${\left\{\left(s,a,r,s^{\prime }\right)\right\}}_{i=1}^{K}$ can be acquired by a proper learning policy; (C2): the reward discount factor γ∈[0,1]; (C3): the learning rate $\delta_{t}$ satisfies*

(35)
$$\begin{array}{c} 0\leq \delta_{t}\leq 1,\sum_{t=1}^{\infty }\delta_{t}=\infty ,\sum_{t=1}^{\infty }{\left(\delta_{t}\right)}^{2}<\infty ; \end{array}$$

*(C4): the reward function $r_{t}$ in the EH-CRN which is defined by the Equation (30) is bounded; (C5): the Q values $Q_{A}$ and $Q_{B}$ are stored in a lookup table, and (C6): both $Q_{A}$ and $Q_{B}$ are updated infinitely many times. Then, Q values $Q_{A}$ and $Q_{B}$ will converge to the optimal value function $Q^{*}$, i.e., $Q_{A},Q_{B}\to Q^{*}$.*


**Proof.** In (C1), the ϵ-greedy policy in the Equation (32) can be used as the proper learning policy to collect a large number of state transitions. In (C2), γ is easily found by taking a value between 0 and 1. In (C3), the learning rate can be set as $\delta_{t}=\frac{1}{t+1}$, and then the formula
(36)$$\begin{array}{c} \left\{\begin{array}{c} 0\leq \frac{1}{t+1}\leq 1,\\ \sum_{t=1}^{\infty }\delta_{t}=\sum_{t=1}^{\infty }\frac{1}{t+1}=\infty ,\\ \sum_{t=1}^{\infty }{\left(\frac{1}{t+1}\right)}^{2}<\sum_{t=1}^{\infty }\frac{1}{t(t+1)}=\sum_{t=1}^{\infty }\left(\frac{1}{t}-\frac{1}{t+1}\right)<1<\infty \end{array}\right. \end{array}$$
holds. In (C4), the reward function $r_{t}$ in the Equation (30) includes three parts: (1) $R_{\sec}\left[t\right]$; (2) $R_{\mathrm{SINR}}\left[t\right]$; (3) $R_{\mathrm{Bac}}\left[t\right]$. For the finite transmission powers $p_{\mathrm{ST}}^{m}$ and $p_{\mathrm{PT}}^{m}$, the finite channel coefficients, the finite battery capacity, and the constant reward weights $\eta_{1},\eta_{2},\eta_{3}$, all parts $R_{\sec}\left[t\right]$, $R_{\mathrm{SINR}}\left[t\right]$ and $R_{\mathrm{Bac}}\left[t\right]$ are also finite, and, thus, the reward function $r_{t}$ are bounded. In (C5), we can create a Q table in the same way as the Q-learning algorithm and then store the Q values $Q_{A}$ and $Q_{B}$ in it. In (C6), $Q_{A}$ and $Q_{B}$ can be updated infinitely by (36) as long as time steps are long enough, i.e., t→∞. Consequently, $Q_{A}$ and $Q_{B}$ can converge to the optimal Q values function $Q^{*}$. The proof is completed. □

## 7. Simulation Results

In this part, we conduct some numerical experiments to verify the proposed DRL-based joint EH time and power allocation scheme for our proposed system. Main simulation parameters are listed in Table 1.

In the simulation setup, the number of multiple sub-carriers is started by N=32. We define the EH circuit parameters $a_{l}=150.0$, $b_{l}=0.014$, $A_{l}=1.5$ W [17]. As shown in Figure 4, the decision Q-network consists of six fully connected layers, an LSTM layer, an output layer, and a Softmax layer. In the LSTM layer, there are five LSTM units, and each unit consists of 100 neurons. The number of neurons in the fully connected layers are 64,64,128,128,128, and 128, respectively. Rectified linear units (ReLUs), which are defined as f(x)=max(0,x), are employed as the activation function of these six fully connected layers. The output layer generates a vector containing the Q-values corresponding to all actions, and then the Softmax layer normalizes the Q-values to zero and one. In the reward, we set the positive weights $\eta_{1}=0.6,\eta_{2}=0.2,\eta_{3}=0.2$, respectively. At the start of each episode, all nodes are distributed randomly at the square area with a 300 m length of a side.

For verifying the performance of our proposed method, our proposed method is compared with the following benchmark schemes for resource allocation.

(1) JOEHTS-TP-QL (Joint Optimization of EH Time-Slot and Transmission Power Based on Q-Learning) in [20]: This method is based on a traditional reinforcement learning algorithm. To apply it to solve our problem, the state space is required to be discretized. It aims to maximize the achievable secrecy rate by optimizing transmission power.

(2) MADDPG Based Resource Allocation in [21]: It aims to maximize achievable secrecy rate by jointly optimizing EH time slot and transmission power.

(3) C-DQN scheme in [22]: This method is a combination of a curiosity-driven mechanism and Deep Q-network (DQN), and the agent is reinforced by an extrinsic reward supplied by the environment and an intrinsic reward.

Figure 5 shows the changes in secrecy rate at each episode during the training phase. In this figure, the secrecy rate under the proposed method represents a growing trend despite of continuous fluctuations during the earlier 1000 episodes and later reaches a convergence at a steady rate, which presents the effectiveness in improving secrecy performance. After 1400 training episodes, the performance gaps between our proposed method and other schemes become distinctive. The MADDPG method shows a wide range of fluctuation and requires more training episodes to converge while our proposed method converges quickly and steadily, and this demonstrates the effectiveness in improving the secrecy rate and overcoming the influence caused by the instability. Compared with the previous schemes, the C-DQN scheme can steadily converge to a lower secrecy rate due to the nonuse of the EH technique. The secrecy rate under the JOEHTS-TP-QL scheme slowly increases and surpasses the C-DQN scheme after 1280 training episodes.

The proposed method solves the basic instability and overestimation problems of using function approximation in reinforcement learning: prioritized experience replay and target network by using deep learning (DL) and reinforcement learning (RL). Prioritized experience replay enables RL agents to sample and train offline from previously observed data. This not only greatly reduces the amount of interaction required by the environment, but also can sample a batch of experiences to reduce the differences in learning and updating. In addition, by uniformly sampling from a large memory, the time dependence that may adversely affect the RL algorithm for RL is broken, thereby improving throughput. Both JOEHTS-TP-QL and C-DQN schemes are not equipped with the above advantages. MADDPG has the following problem: each critic network needs to observe the state and action of all agents, and it is not particularly practical for a large number of uncertain agent scenarios, and when the number of agents is too large, the state space is too large. Based on this, they fall behind the proposed method.

Figure 6 and Figure 7 show the secrecy rates with respect to increasing transmission powers of ST and jammer under different schemes, respectively. In Figure 6, the transmission power of the jammer is fixed as 2.0 W. The secrecy rate under our proposed method increases as the maximum transmission power of ST becomes greater, and the maximum secrecy rate is obtained when it is between 1.5 W and 2.5 W. Increasing the maximum transmission power contributes to a greater secrecy rate in some ways, but a greater transmission power is likely to cause strong interferences to the PU and, thus, jeopardize the spectrum access of the SU for gaining further performance enhancement. The C-DQN and the JOEHTS-TP-QL schemes have similarly low secrecy rates. In Figure 7, the maximum transmission power of ST is fixed as 2.0 W. It can be observed that when the maximum transmission power of the jammer is kept below a certain power level, the jamming signal of the jammer has little impact on the eavesdropper. This indicates that the jammer must harvest enough energy to increase the transmission power such that the secrecy performance can be improved further.

We study the effect of different discount rates on the cumulative reward per episode. In Figure 8, the more the discount rate deviates from 1.0, the more dramatically the trajectory of the reward fluctuates. When γ is set as 1.0, the reward converges in the fastest and most stable way. Theoretically, the agent is likely to focus on short-term returns when the discount rate γ is lower than 1.0. During the training process, the secrecy rate as a main part of the reward dominates the learning direction of the agent, hence maximizing the cumulative rewards encourages more instantaneous secrecy information to be delivered. However, agents cannot transmit secrecy data for long periods of time due to the limited energy. Therefore, γ=1.0 acts as an optimal balance factor between the EH and WIT phases.

Figure 9 and Figure 10 show the secrecy rates under all the algorithms versus the maximum battery capacity of jammer and ST, respectively. With the increase in the maximum battery capacity, the secrecy rates under different schemes also increase. Of all the algorithms, the proposed algorithm gains the best performance at each given value of maximum battery capacity. When the maximum battery capacity is beyond 60 mJ, all algorithms start to converge to different performance levels; this is mainly because the jammer and ST are influenced by the limited RF energy that can be harvested.

Figure 11 shows the secrecy rate versus the number of subcarriers *m*. The proposed method is capable of greatly enhancing the secrecy rate and outperforms other schemes with the highest secrecy rate. The MADDPG and the C-DQN schemes have performance gaps of approximately 28.8% and 80.5% with our proposed method, respectively, when the number of subcarriers is 128. The secrecy rate under the JOEHTS-TP-QL scheme degrades dramatically when the number of subcarriers is beyond 96. It is observed that the secrecy rates under all schemes start to deteriorate when the number of subcarriers is beyond 128. It is mainly because increasing the number of subcarriers will increase the size of the action space, and it is harder to find the optimal strategy, which brings a decline in performance.

With the knowledge of the environment regularity, agents can intelligently adjust their decision strategies so that a target state can be obtained from any initial state in a few numbers of transition steps. Here, a target state is defined as a given state where all constraint requirements in the proposed problem are satisfied. Similar to [27], we use the indicator “average number of transition steps”, which is defined as the average number of continuous transition steps agents take from an initial state to a target state, to measure the robustness performance for our proposed algorithm.

Figure 12 shows the average number of transition steps in each test. Multi-agents are tested in five hundred time steps at the end of each training episode. Our proposed scheme only requires the smallest number of transition steps to achieve the target state, while the other schemes need to take more transition steps. Moreover, when it is the 25th test, the proposed method fundamentally converges while other schemes at least need to take 65 tests and even more, and, thus, this makes the convergence speed of the proposed method 160% faster than the benchmark algorithm.

This corroborates the robustness and rapidity of the proposed method.

## 8. Conclusions

In this paper, we have developed a multi-agent DRL framework for the proposed EH-based CRN with a wireless-powered cooperative jammer and propose the corresponding resource allocation problem. The D3QN algorithm is combined with an LSTM network to improve the system’s secrecy performance. The proposed method is divided into training and implementation phases. The numerical results demonstrate that the proposed method can increase the long-term achievable secrecy rate by 30.1% and convergence speed by 160% with the minimum average number of transition steps overheads, compared with the benchmark algorithms. In the future study, we will further explore the secrecy of energy-efficient resource allocation for our proposed network. 

## Figures and Tables

**Figure 1 sensors-23-00807-f001:**
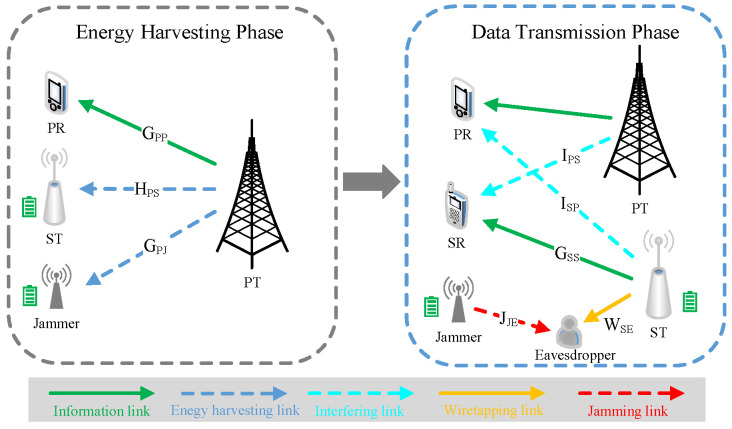
A CRN structure.

**Figure 2 sensors-23-00807-f002:**
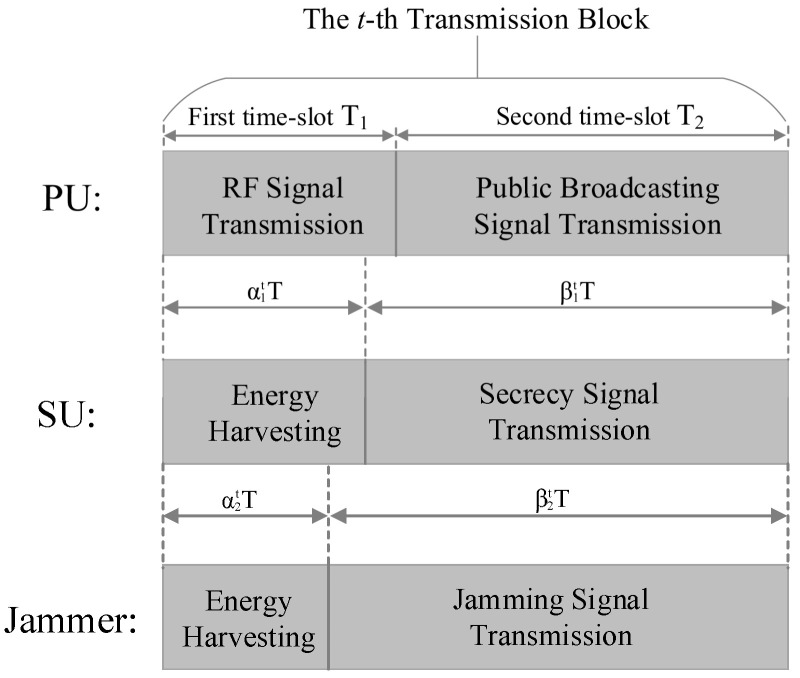
A two-phase EH transmission scheme.

**Figure 3 sensors-23-00807-f003:**
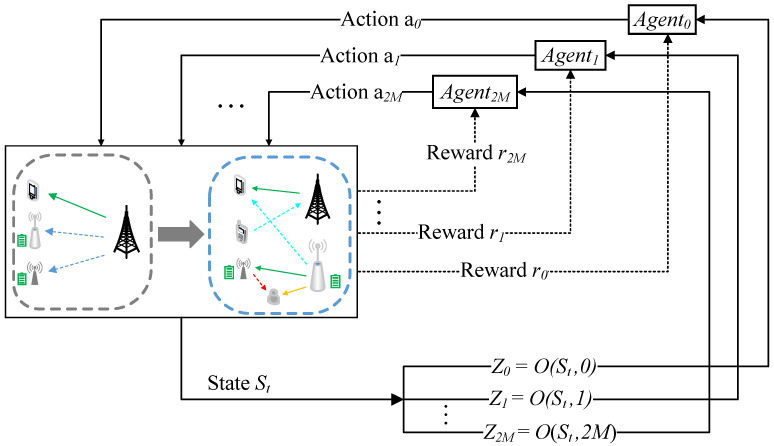
A DRL framework.

**Figure 4 sensors-23-00807-f004:**
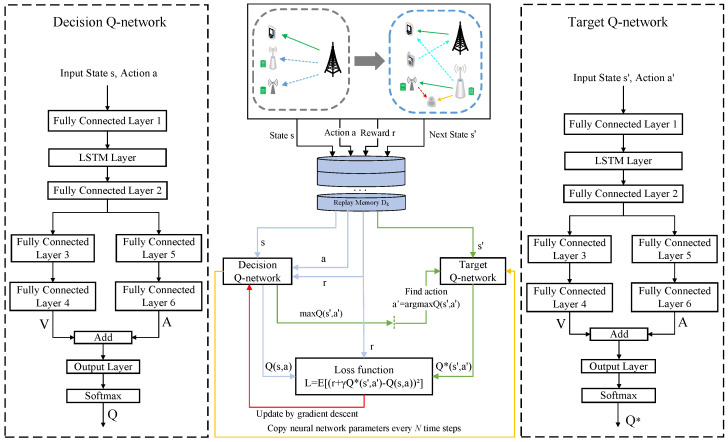
LSTM-D3QN network architecture.

**Figure 5 sensors-23-00807-f005:**
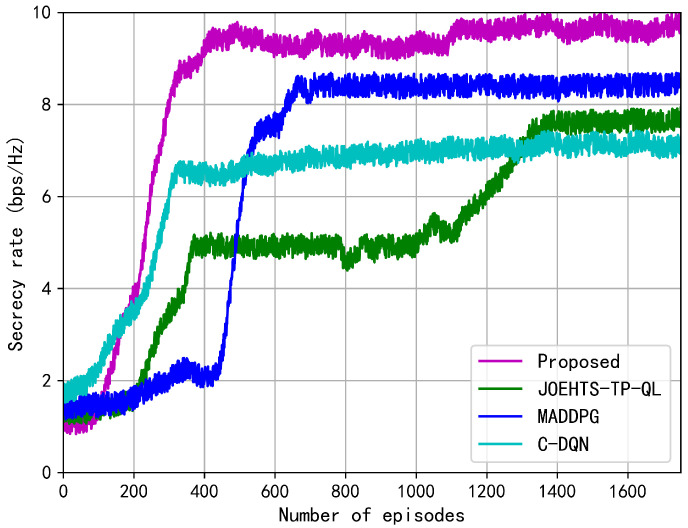
Secrecy rate versus the number of episodes.

**Figure 6 sensors-23-00807-f006:**
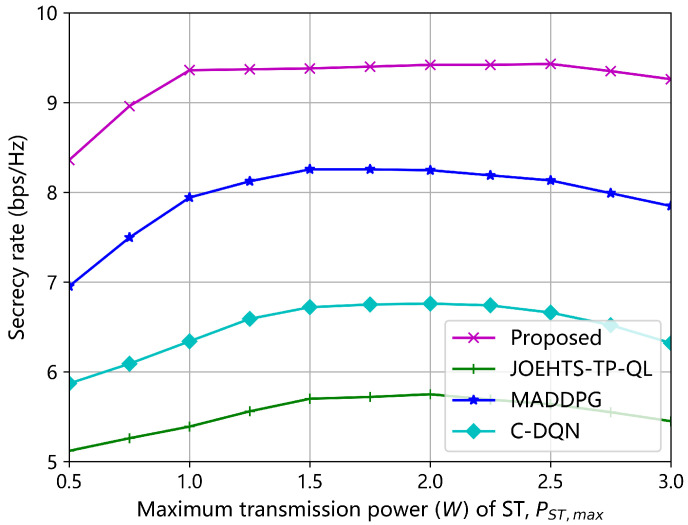
Secrecy rate versus maximum transmission power of ST.

**Figure 7 sensors-23-00807-f007:**
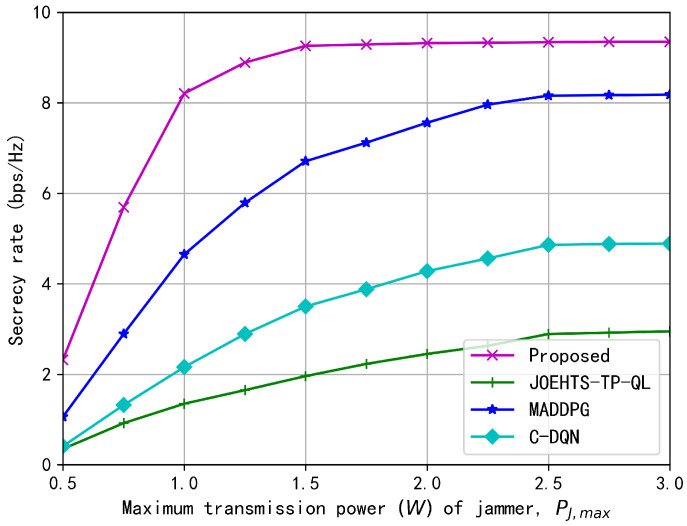
Secrecy rate versus maximum transmission power of jammer.

**Figure 8 sensors-23-00807-f008:**
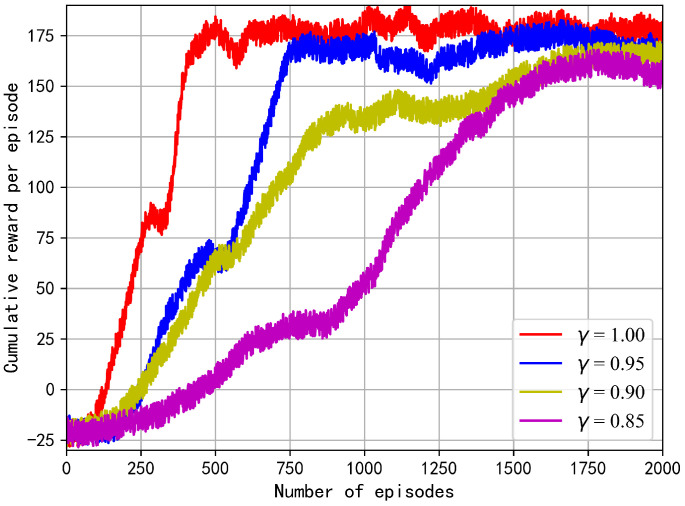
Reward under different discount rates.

**Figure 9 sensors-23-00807-f009:**
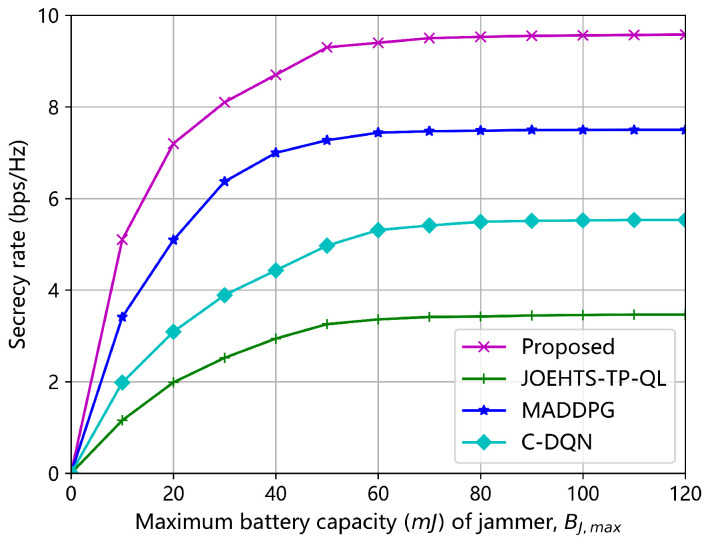
Secrecy rate versus maximum battery capacity $B_{J,\max}$ of jammer.

**Figure 10 sensors-23-00807-f010:**
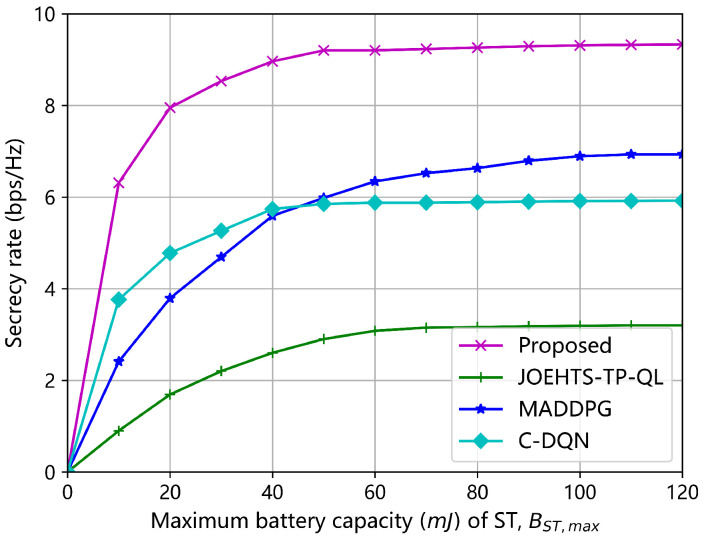
Secrecy rate versus maximum battery capacity $B_{\mathrm{ST},\max}$ of ST.

**Figure 11 sensors-23-00807-f011:**
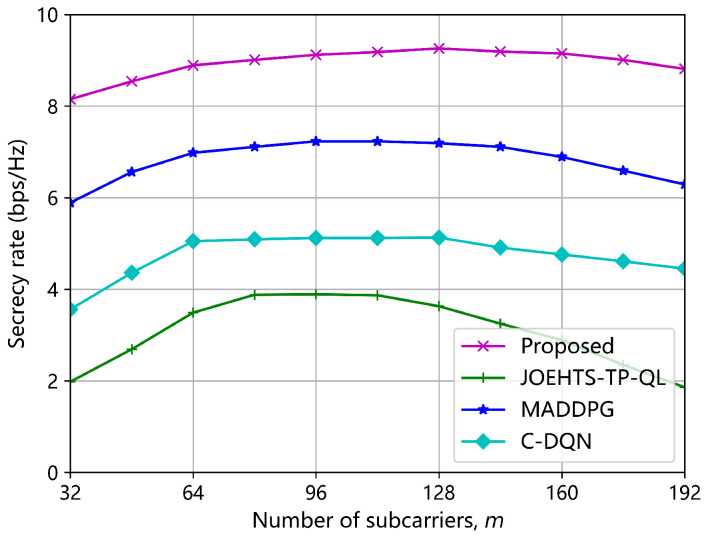
Secrecy rate versus the number of subcarriers *m*.

**Figure 12 sensors-23-00807-f012:**
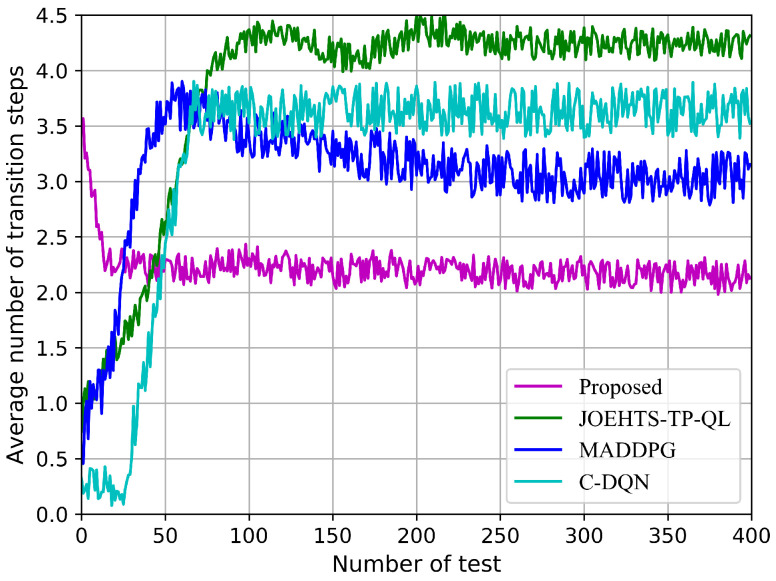
Average number of transition steps in each test.

**Table 1 sensors-23-00807-t001:** Main Simulation Parameters.

Parameter	Value
Carrier frequency	750 MHz
Bandwidth per channel	10 MHz
Number of training episodes	2000
Number of training steps per episode	1000
Number of LSTM units	5
Number of neurons of an LSTM unit	100
Number of sub-carriers	32
Noise power $\sigma^{2}$	0.0001 W
Transmission powers for the PT	[1.0,1.5,2.0,2.5,3.0] W
Transmission powers for the ST	[0.5,1.0,1.5,2.0,2.5,3.0] W
Transmission powers for the jammer	[0.5,1.0,1.5,2.0,2.5,3.0] W
Learning rate (δ)	0.00025
Discount rate (γ)	0.95
Initial exploration rate	0.99

## Data Availability

Not applicable.

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
