# Peer review of "Deep Reinforcement Learning for Physical Layer Security Enhancement in Energy Harvesting Based Cognitive Radio Networks"

_sensors, 2023, doi:10.3390/s23020807_

Round 1

Reviewer 1 Report

This paper consider the secrecy communication in EH-based cognitive radio networks with a cooperative jammer. Based on the assumptions about jamming signals and energy harvesting, a multi-agent DRL method is proposed to optimize the power allocation and time allocation. Simulation results verify the effectiveness of the proposed scheme.

The reviewer has the following concerns:

1.      P4 line 175, “each node is equipped with a single antenna”, P5 line 193, the jamming signal is complex Gaussian random variable. From Equ. (6) and (7), it is apparently that PR and SR can eliminate the jamming signals. Based on the knowledge that each node has only single antenna and the jamming signal is random, how can PR and SR cancel the jamming signals? This is critical since the secrecy of the scheme is based on this assumption.

2.      From Rqu (6) and (7) we can see that SR can receive the signals transmitted by PT which is S_{PT}^m, why the eavesdropper can not receive the primary signal S_{PT}^m?

3.      P5, the transmission scheme has two phases. In EH phase, the PT broadcasts wireless RF energy signals, in Phase 2, PU updates its transmission power. If the time durations for Phase 1 and 2 are T1 and T2, then T1+T2=T. The portions allocated for EH at ST and Jammer are respectively \alpha_1^t and \alpha_2^t which satisfy the constraints in Equ. (1) and (2). It is not clear that in Phase 2, PU still transmits the RF energy signal or transmits the primary signals. If T\alpha_1^t >T1, for the duration T\alpha_1^t -T1, can ST still harvest energy from the PU’s transmission? If the PU transmissions in Phase 1 and Phase 2 are different, the energy harvested in durations T1 and T2 should be different, then the energy harvested and the time portion should be considered jointly.

4.      P8, Section 5.2, the instantaneous CSIs from different links including the CSI of the SE link and the JE link are known at each agent. The eavesdropper is passive and noncooperative, how can the agents get the instantaneous CSIs of the SE link and the JE link?

5.      In this paper each subcarrier is models as an agent, the agents share the same information and the same reward. Why it is necessary to use multi-agent DRL which is a distributed algorithm? Why not use reinforcement learning?

Author Response

Response to Review of Paper:

Manuscript ID: sensors-2110570

Title: Deep Reinforcement Learning for Physical Layer Security Enhancement in Energy Harvesting Based Cognitive Radio Networks

Dear Editor and Reviewers,

We would like to express our sincere thanks to you for your thorough and insightful comments. We have tried our best to revise the paper by seriously addressing each individual issue raised. In the following, we present a point-to-point response to each individual comment and elaborate on how the manuscript has been revised. We have uploaded the revised papers with and without revision marks as well. Again, we sincerely appreciate your time on this paper.

Best Regards,

Ruiquan Lin, Hangding Qiu, Weibin Jiang, Zhenglong Jiang, Zhili Li and Jun Wang

AUTHORS’ RESPONSES TO REVIEWER 1

Comment 1:

P4 line 175, “each node is equipped with a single antenna”, P5 line 193, the jamming signal is complex Gaussian random variable. From Equ. (6) and (7), it is apparently that PR and SR can eliminate the jamming signals. Based on the knowledge that each node has only single antenna and the jamming signal is random, how can PR and SR cancel the jamming signals? This is critical since the secrecy of the scheme is based on this assumption

Reply:   Thank you for your astute comments. We have added the explanation about this assumption in Section 5.1 as follows:

"… … This can be realized by the following method: A large number of random sequences (jamming signals) with Gaussian distribution are pre-stored at the jammer and their indices are the keys. The jammer randomly selects a sequence (jamming signal) and sends its key to the SR. The key can be sent in a secret manner via channel independence and reciprocity. As the random sequence is only known at the SR, any potential eavesdropper cannot access the random sequence."

Comment 2:

From Rqu (6) and (7) we can see that SR can receive the signals transmitted by PT which is S_{PT}^m, why the eavesdropper can not receive the primary signal S_{PT}^m?

Reply:   Thanks for the reviewer’s comment. In the version submitted previously, on this point, we make the following explanation:

In this paper, the signal transmitted by the primary user can be a public broadcast signal, which is visible to all users. Eavesdropper can receive the signal of the primary user, but such signal is meaningless to eavesdropper, because all users can receive such signal. What we study in this paper is the secrecy communication of the secure user network. We mainly consider the secrecy signals transmitted by secure users, therefore, the signal received by the eavesdropper only includes the secure user's secret signal.

Considering the suggestion of reviewers, we have added relevant explanations in the System model part in the revised version of the paper, as follows:

In the second time slot, namely the data transmission phase, the PU instantaneously updates its transmission power based on a power control strategy but is unknown to the SU [29], and transmits the public broadcasting signals to the PR, because the public broadcasting signal is meaningless to eavesdroppers, this paper only considers that the signal received by the eavesdroppers only includes the secrecy signal of SU.

Comment 3:

P5, the transmission scheme has two phases. In EH phase, the PT broadcasts wireless RF energy signals, in Phase 2, PU updates its transmission power. If the time durations for Phase 1 and 2 are T1 and T2, then T1+T2=T. The portions allocated for EH at ST and Jammer are respectively \alpha_1^t and \alpha_2^t which satisfy the constraints in Equ. (1) and (2). It is not clear that in Phase 2, PU still transmits the RF energy signal or transmits the primary signals. If T\alpha_1^t >T1, for the duration T\alpha_1^t -T1, can ST still harvest energy from the PU’s transmission? If the PU transmissions in Phase 1 and Phase 2 are different, the energy harvested in durations T1 and T2 should be different, then the energy harvested and the time portion should be considered jointly.

Reply:   Thanks for your kind comments. We make the following explanations about this:

Similar to this literature [Intelligent Power Control for Spectrum Sharing in Cognitive Radios: In A Deep Reinforcement Learning Approach]: at the beginning of each time slot, the primary user immediately updates the power, while the secondary user synchronously receives the spectrum of the primary user, so this is a synchronization process of the primary and secondary users. We draw on this literature, and we also consider the synchronization process of primary and secondary users. We have supplemented the following contents in the System Model part of the revised paper, as follows:

“In the second time slot, namely the data transmission phase, the PU instantaneously updates its transmission power based on a power control strategy but is unknown to the SU \cite{ref29}, and transmits the public broadcasting signals to the PR, … …

“Similar to [29], the information transmissions of PU and SU in the above two phases is considered as a synchronization process.”

Comment 4:

P8, Section 5.2, the instantaneous CSIs from different links including the CSI of the SE link and the JE link are known at each agent. The eavesdropper is passive and noncooperative, how can the agents get the instantaneous CSIs of the SE link and the JE link?

Reply: Thanks for your kind reminder. In the Introduction part of revised version of the paper, we have added an explanation that the revised paper only considers the situation that the eavesdropper's location information and channel state information cannot be accurately obtained:

“The research on physical layer security is usually divided into two cases: one is that the Channel State Information (CSI) of the eavesdropper is known, the other is that the channel state information is not perfect. In most practical cases, the accurate location and CSI of the eavesdropper are unknown to the network. Our work considers the second case, which is a common assumption in the field of physical layer security.”

Much of the literature makes this assumption, such as literatures [Security-aware Max -- Min Resource Allocation in Multiuser OFDMA Downlink], [Secrecy Performance of Underlay Cognitive Radio Networks With Primary Interference], [Secure and Energy Efficient Transmission for IRS-Assisted Cognitive Radio Networks] and so on. Therefore, based on their achievements, we also directly make such an assumption.

Comment 5:

In this paper each subcarrier is models as an agent, the agents share the same information and the same reward. Why it is necessary to use multi-agent DRL which is a distributed algorithm? Why not use reinforcement learning?

Reply: Thanks for your kind reminder. We make the following explanations about this:

We take the channel state information of the system as a part of the state, which makes the state space of the system very large. Reinforcement learning algorithm like Q learning cannot store such a large state value, and it cannot solve the Markov Decision Process under such an environment. However, deep neural network can easily fit nonlinear functions. The state value can be solved by the nonlinear function, so this paper adopts the deep reinforcement learning (DRL) algorithm instead of the reinforcement learning algorithm.

We have added the following content in the related work part, as follows:

… … Q-Learning algorithm is a typical model-free algorithm, the update of its Q table is different from the strategy followed when selecting actions, in other words, when updating the Q table, the maximum value of the next state is calculated, but the action corresponding to the maximum value does not depend on the current strategy. However, Q-learning algorithm faces two main problems: 1) when the state space is large, Q-learning will consume a lot of time and space for searching and storage. (2) Q-learning has the problem of overestimation for state-action value. Considering the fatal shortcomings of Q-learning algorithm, we further turn to the research of DRL algorithm.”

We have added the following content about the strong points of DRL in the Simulation Results part, as follows:

“The proposed method solves the basic instability and overestimation problems of using function approximation in reinforcement learning: prioritized experience replay and target network by using deep learning (DL) and reinforcement learning (RL). Prioritized experience replay enables RL agents to sample and train offline from previously observed data. This not only greatly reduces the amount of interaction required by the environment, but also can sample a batch of experiences to reduce the differences in learning and updating. In addition, by uniformly sampling from a large memory, the time dependence that may adversely affect the RL algorithm for RL is broken, thereby improving throughput.”

Regards,

Ruiquan Lin, Hangding Qiu, Weibin Jiang, Zhenglong Jiang, Zhili Li and Jun Wang

Reviewer 2 Report

In this paper, the authors proposed a multi-agent deep reinforcement learning (DRL) method for solving the optimization of resource allocation and performance. Overall, the authors have made a good attempt. However, the reviewer has the following comments:

1.      The similarity of this paper is high. In iTheticate, the similarity is 33%. The authors should improve the presentation.

2.      In abstract, the result of this work must be described briefly. The result of this work is not clear. The authors only described that “The simulation results show that the proposed method is efficient in improving the system secrecy performance, as compared to other benchmark schemes”. What do the “other benchmark schemes” stand for?

3.      The problem definition of this work is not clear. In Sect. 2, the drawbacks of each conventional technique should be described one by one. The authors should emphasize the difference with other methods to clarify the position of this work further.

4.      In the Introduction part, strong points of this proposed method should be further stated and organization of this whole paper is supposed to be provided in the end.

5.      This paper lacks in-depth discussions in Sect. 6. The impact is lost by a short discussion of the findings. Readers will fail to understand the scientific contribution of this research. Show the theoretical reason why the proposed technique is better than the existing techniques [32, 33], because there is no theoretical explanation about compared existing techniques in previous sections. These existing techniques appeared suddenly in comparison. Explain the detail of the existing technique in previous sections.

6.      In section [6], the effectiveness of this work is not clear. Through simulations, the authors must justify the effectiveness of the proposed method by comparing with the state-of-the-art methods. Several articles are discussed in the research survey. However, no comparison is shown with these techniques. Frankly speaking, the research survey and References are meaningless. The authors should show comparison data.

7.      The results of this research are not clear in Conclusions. Show the scientific contribution of this work with concrete data.

Author Response

Response to Review of Paper:

Manuscript ID: sensors-2110570

Title: Deep Reinforcement Learning for Physical Layer Security Enhancement in Energy Harvesting Based Cognitive Radio Networks

Dear Editor and Reviewers,

We would like to express our sincere thanks to you for your thorough and insightful comments. We have tried our best to revise the paper by seriously addressing each individual issue raised. In the following, we present a point-to-point response to each individual comment and elaborate on how the manuscript has been revised. We have uploaded the revised papers with and without revision marks as well. Again, we sincerely appreciate your time on this paper.

Best Regards,

Ruiquan Lin, Hangding Qiu, Weibin Jiang, Zhenglong Jiang, Zhili Li and Jun Wang

AUTHORS’ RESPONSES TO REVIEWER 2

Comment 1:

The similarity of this paper is high. In iTheticate, the similarity is 33%. The authors should improve the presentation.

Reply:   Thanks for your astute comments. We have improved our presentation and the similarity in the revised paper is 14.6%.

Comment 2:

In abstract, the result of this work must be described briefly. The result of this work is not clear. The authors only described that “The simulation results show that the proposed method is efficient in improving the system secrecy performance, as compared to other benchmark schemes”. What do the “other benchmark schemes” stand for?

Reply:   Thanks for the reviewer’s suggestion. In abstract, we have revised the presentation about the result of this work, as follows:

"… … The simulation results show that the proposed method outperforms the benchmark schemes in terms of secrecy rate, convergence speed and average number of transition steps. "

“other benchmark schemes” stand for the three compared methods in the simulation part.

“1) JOEHTS-TP-QL (Joint Optimization of EH Time-Slot and Transmission Power Based on Q-Learning) in [20]: This method is based on a traditional reinforcement learning algorithm. In order to apply it to solve our problem, the state space is required to be discretized. It aims to maximize achievable secrecy rate by optimizing transmission power.

2) MADDPG Based Resource Allocation in [21]: It aims to maximize achievable secrecy rate by joinly optimizing EH time slot and transmission power.

3) C-DQN scheme in [22]: This method is a combination of a curiosity-driven mechanism and Deep Q-network (DQN), and the agent is reinforced by an extrinsic reward supplied by the environment and an intrinsic reward.”

Comment 3:

The problem definition of this work is not clear. In Sect. 2, the drawbacks of each conventional technique should be described one by one. The authors should emphasize the difference with other methods to clarify the position of this work further.

Reply:   Thanks for your astute comments. In Sect. 2, We have described the drawbacks of each conventional technique and emphasized the difference with other methods as follows:

… … Linear EH in [13],[14],[15] is an ideal acquisition model while non-linear EH model can better reflect the actual situation of EH.”

“The proposed MADDPG method is an extension of DDPG in multi-agent tasks. Its basic idea is centralized learning and decentralized execution. During training, the actor network is updated by a global critical network, while during testing, the actor network takes actions. MADDPG has a distinct advantage in dealing with the continuous state and action spaces, compared with the traditional reinforcement learning method in [20]. However, the MADDPG algorithm is also susceptible to large state space, e,g, the large state space easily lead an unstable convergence process.”

“On the one hand, C-DQN algorithm uses deep convolutional neural network to approximate the state-action value function, so the model has a better robustness, on the other hand, C-DQN also has the same overestimation problem as Q-learning algorithm.”

“In the above literature, although Q-learning algorithm in [20] and C-DQN algorithm in [22] can improve the system performance to some extent, there exists the overestimation of the state-action value. There is an urgent need for an improved DRL algorithm that can overcome the overestimation and improve system performance to the greatest extent.”

Comment 4:

 In the Introduction part, strong points of this proposed method should be further stated and organization of this whole paper is supposed to be provided in the end.

Reply: Thank you for your astute comments. We have further added strong points of this proposed method in the main contributions part and given the organization of this whole paper in the Paper Organization part:

“Besides, the construction of loss function in the proposed method differently from previously mentioned algorithms, and thus it can well avoid the overestimation of the state-action value and make the training process more stable”

“The remainder of the paper is organized as follows. Section 5 presents an EH-based CRN system model and a related optimization problem. Section 6 proposes a multi-agent DRL framework to obtain a solution to this joint EH time and power allocation problem. Section 7 presents simulation results to evaluate the system performance via our proposed method as compared to benchmark schemes. Finally, Section 8 concludes this paper.”

Comment 5:

This paper lacks in-depth discussions in Sect. 6. The impact is lost by a short discussion of the findings. Readers will fail to understand the scientific contribution of this research. Show the theoretical reason why the proposed technique is better than the existing techniques [32, 33], because there is no theoretical explanation about compared existing techniques in previous sections. These existing techniques appeared suddenly in comparison. Explain the detail of the existing technique in previous sections.

Reply:   Thank you for your astute comments. We have shown the theoretical reason why the proposed technique is better than the existing techniques as follows:

“The proposed method solves the basic instability and overestimation problems of using function approximation in reinforcement learning: prioritized experience replay and target network by using deep learning (DL) and reinforcement learning (RL). Prioritized experience replay enables RL agents to sample and train offline from previously observed data. This not only greatly reduces the amount of interaction required by the environment, but also can sample a batch of experiences to reduce the differences in learning and updating. In addition, by uniformly sampling from a large memory, the time dependence that may adversely affect the RL algorithm for RL is broken, thereby improving throughput. Both JOEHTS-TP-QL and C-DQN schemes are not equipped with the above advantages. MADDPG has the following problem: each critic network needs to observe the state and action of all agent, and it is not particularly practical for a large number of uncertain agent scenarios, and when the number of agent is too large, the state space is too large. Based on this, they fall behind the proposed method.”

We have added a detailed explanation of the three compared methods in the introduction part as follows:

“Q-Learning algorithm is a typical model-free algorithm, the update of its Q table is different from the strategy followed when selecting actions, in other words, when updating the Q table, the maximum value of the next state is calculated, but the action corresponding to the maximum value does not depend on the current strategy. However, Q-learning algorithm faces two main problems: 1) when the state space is large, Q-learning will consume a lot of time and space for searching and storage. (2) Q-learning has the problem of overestimation for state-action value.”

“The proposed MADDPG method is an extension of DDPG in multi-agent tasks. Its basic idea is centralized learning and decentralized execution. During training, the actor network is updated by a global critical network, while during testing, the actor network takes actions. MADDPG has a distinct advantage in dealing with the continuous state and action spaces, compared with the traditional reinforcement learning method in [20]. However, the MADDPG algorithm is also susceptible to large state space, e,g, the large state space easily lead an unstable convergence process.”

“On the one hand, C-DQN algorithm uses deep convolutional neural network to approximate the state-action value function, so the model has a better robustness, on the other hand, C-DQN also has the same overestimation problem as Q-learning algorithm.”

Comment 6:

In section [6], the effectiveness of this work is not clear. Through simulations, the authors must justify the effectiveness of the proposed method by comparing with the state-of-the-art methods. Several articles are discussed in the research survey. However, no comparison is shown with these techniques. Frankly speaking, the research survey and References are meaningless. The authors should show comparison data.

Reply:   Thanks for your astute comments. We have replaced the compared method as that of references [20], [21], [22] and shown comparison data in the Simulation results part.

“1)JOEHTS-TP-QL (Joint Optimization of EH Time-Slot and Transmission Power Based on Q-Learning)} in [20]: This method is based on a traditional reinforcement learning algorithm. In order to apply it to solve our problem, the state space is required to be discretized. It aims to maximize achievable secrecy rate by optimizing transmission power.

2) MADDPG Based Resource Allocation} in [21]: It aims to maximize achievable secrecy rate by joinly optimizing EH time slot and transmission power.

3) C-DQN scheme in [22]: This method is a combination of a curiosity-driven mechanism and Deep Q-network (DQN), and the agent is reinforced by an extrinsic reward supplied by the environment and an intrinsic reward.”

“The MADDPG and the C-DQN schemes have performance gaps of approximate 28.8% and 80.5% with our proposed method respectively when the number of subcarriers is 128.”

“Besides, when it is the 25-th test, the proposed method fundamentally converges while other schemes at least need to take 65 tests and even more, and thus this makes the convergence speed of the proposed method 160% faster than the benchmark algorithm.”

“The numerical results demonstrate that the proposed method can increase the long-term achievable secrecy rate by 30.1% and convergence speed by 160% with minimum average number of transition steps overheads, compared with the benchmark algorithms.”

Comment 7:

The results of this research are not clear in Conclusions. Show the scientific contribution of this work with concrete data.

Reply:   Thanks for your astute comments. We have described the results of this research and the concrete data in Conclusions part:

… … The numerical results demonstrate that the proposed method can increase the long-term achievable secrecy rate by 30.1% and convergence speed by 160% with minimum average number of transition steps overheads, compared with the benchmark algorithms.”

Regards,

Ruiquan Lin, Hangding Qiu, Weibin Jiang, Zhenglong Jiang, Zhili Li and Jun Wang

Round 2

Reviewer 1 Report

The authors have answered some of my concerns.
However, more explanations are necessary for some questions.

For example, Comment 3, In Phase I, the wireless RF signals are transmitted which is exploited by SU and the jammer to harvest energy as in (3) (4). In Phase II PU broadcast public data signals with a different power level (unknown to SU). The time duration for Phase I and Phase II is T, and \alpha_1^t \times T+\beta_1^t \times T=T (Equ. 1 in the paper). Only when \alpha_1^t \times T is smaller than the duration of Phase I (denoted as T_1), Equ. (3) is correct. That is why I ask the authors in Comment 3 “If T\alpha_1^t >T1, for the duration T\alpha_1^t -T1, can ST still harvest energy from the PU’s transmission? If the PU transmissions in Phase 1 and Phase 2 are different, the energy harvested in durations T1 and T2 should be different, then the energy harvested and the time portion should be considered jointly.” But the authors only emphasized that PU and SU are synchronized. The concerns of Comment 3 is not about synchronization, it is about how much energy can be harvested. This is quite critical since the following algorithm is based on Equ. (3) and (4).

Beside, this paper has some quite strong assumptions. For example, in the reply to Comment 1, it is assumed some key can be shared in a secret manner via channel independence and reciprocity. If it is true, there are many other ways to provide secrecy other than transmitting jamming signals at the cost of extra power consumption. 

Author Response

Response to Review of Paper:

Manuscript ID: sensors-2110570

Title: Deep Reinforcement Learning for Physical Layer Security Enhancement in Energy Harvesting Based Cognitive Radio Networks

Dear Editor and Reviewers,

We would like to express our sincere thanks to you for your thorough and insightful comments. We have tried our best to revise the paper by seriously addressing each individual issue raised. In the following, we present a point-to-point response to each individual comment and elaborate on how the manuscript has been revised. We have uploaded the revised papers with and without revision marks as well. Again, we sincerely appreciate your time on this paper.

Best Regards,

Ruiquan Lin, Hangding Qiu, Weibin Jiang, Zhenglong Jiang, Zhili Li and Jun Wang

AUTHORS’ RESPONSES TO REVIEWER 1

Comment 1:

For example, Comment 3, In Phase I, the wireless RF signals are transmitted which is exploited by SU and the jammer to harvest energy as in (3) (4). In Phase II PU broadcast public data signals with a different power level (unknown to SU). The time duration for Phase I and Phase II is T, and \alpha_1^t \times T+\beta_1^t \times T=T (Equ. 1 in the paper). Only when \alpha_1^t \times T is smaller than the duration of Phase I (denoted as T_1), Equ. (3) is correct. That is why I ask the authors in Comment 3 “If T\alpha_1^t >T1, for the duration T\alpha_1^t -T1, can ST still harvest energy from the PU’s transmission? If the PU transmissions in Phase 1 and Phase 2 are different, the energy harvested in durations T1 and T2 should be different, then the energy harvested and the time portion should be considered jointly.” But the authors only emphasized that PU and SU are synchronized. The concerns of Comment 3 is not about synchronization, it is about how much energy can be harvested. This is quite critical since the following algorithm is based on Equ. (3) and (4).

Reply:   Thanks for reviewers’ kind comments. In the first time-slot, the PU broadcasts the RF signals, and in the meanwhile, the SU harvests and store the energy from the RF signals transmitted by the PU. In the second time slot, the PU transmits the public data signal, and the SU no longer harvest energy from the PU’s transmissions but performs its own secrecy data transmission. Therefore, for the previous Comment 3 “If T\alpha_1^t >T1, for the duration T\alpha_1^t -T1, can ST still harvest energy from the PU’s transmission? If the PU transmissions in Phase 1 and Phase 2 are different, the energy harvested in durations T1 and T2 should be different, then the energy harvested and the time portion should be considered jointly.”, the SU can only harvest energy from the RF signal of PU, so the energy harvested by the SU is determined by the first time slot, which means that "T\alpha_1^t <T1” and "T\alpha_2^t <T1” holds.

Based on this, we performed corresponding modifications in the revised version of the paper:

In the System Model part, we added corresponding constraints to Equ. (3) and (4) to ensure that the harvested energy of SU is from the RF signal of PU, rather than the public broadcasting signal of PU in the second time slot.

Comment 2:

Beside, this paper has some quite strong assumptions. For example, in the reply to Comment 1, it is assumed some key can be shared in a secret manner via channel independence and reciprocity. If it is true, there are many other ways to provide secrecy other than transmitting jamming signals at the cost of extra power consumption.

Reply: Thanks for reviewers’ kind comments. Indeed, there are many other ways to provide secrecy other than transmitting jamming signals at the cost of extra power consumption, such as Beamforming and Precoding technologies in Multi antennas transmissions. These methods can be used as anti-eavesdropping methods, but their application scenarios are different with Cooperative Jammer (CJ) we use. We provided the reason for using CJ method and emphasized its advantage in the Introduction part, as follows:

 “In addition, there are also beamforming and precoding secure transmission methods, however, the complexity of the Beamforming and Precoding schemes in the actual wireless communication system is critical to the operation of the system, and the extremely high computational complexity makes it difficult to apply it in practical systems. In the research of existing papers, CJ is one of the most important ways to achieve secure PLS transmission. In CJ secure transmission scheme, the jammer can complete the design of jamming signals beamforming vector by using the statistical Channel State Information (CSI) of illegal channels, which is more suitable for actual wireless communication scenarios. With this in mind, in this paper, we apply the CJ method to our proposed network.”

Thanks again for reviewers’ comments.

LIST OF MAJOR MODIFICATIONS:

WHERE

Content

Reason

Page 7

Section 5.1

Equ. (4)

Add constraint

Comment #1 of Reviewer #1

Page 6

Section 5.1

Equ. (3)

Add constraint

Comment #1 of Reviewer #1

Page 6

Section 5.1

Add the Figure 2.

Comment #1 of Reviewer #1

Page 2

Introduction

Para. 3

Line 57-66

Add “In addition, there are also beamforming and precoding secure transmission methods, … … , in this paper, we apply the CJ method to our proposed network.”

Comment #2 of Reviewer #1

Regards,

Ruiquan Lin, Hangding Qiu, Weibin Jiang, Zhenglong Jiang, Zhili Li and Jun Wang

Reviewer 2 Report

In this paper, the authors proposed a multi-agent deep reinforcement learning (DRL) method for solving the optimization of resource allocation and performance. In the revised version, all reviewer’s requests were met by the authors. The reviewer would like to pay tribute to the authors’ great work. This is well written and organized paper. It is scientifically sound and contains sufficient interest to merit publication, I think.

Author Response

Response to Review of Paper:

Manuscript ID: sensors-2110570

Title: Deep Reinforcement Learning for Physical Layer Security Enhancement in Energy Harvesting Based Cognitive Radio Networks

Dear Editor and Reviewers,

We would like to express our sincere thanks to you for your thorough and insightful comments. We have tried our best to revise the paper by seriously addressing each individual issue raised. In the following, we present a point-to-point response to each individual comment and elaborate on how the manuscript has been revised. We have uploaded the revised papers with and without revision marks as well. Again, we sincerely appreciate your time on this paper.

Best Regards,

Ruiquan Lin, Hangding Qiu, Weibin Jiang, Zhenglong Jiang, Zhili Li and Jun Wang

AUTHORS’ RESPONSES TO REVIEWER 2

Comment 1:

In this paper, the authors proposed a multi-agent deep reinforcement learning (DRL) method for solving the optimization of resource allocation and performance. In the revised version, all reviewer’s requests were met by the authors. The reviewer would like to pay tribute to the authors’ great work. This is well written and organized paper. It is scientifically sound and contains sufficient interest to merit publication, I think.

Reply:   Thank you for your approval of this paper. We benefited a lot from your suggestions.

Regards,

Ruiquan Lin, Hangding Qiu, Weibin Jiang, Zhenglong Jiang, Zhili Li and Jun Wang

Round 3

Reviewer 1 Report

I have no more comments